# Comparison of Transperitoneal and Retroperitoneal Robotic Partial Nephrectomy for Patients with Completely Lower Pole Renal Tumors

**DOI:** 10.3390/jcm12020722

**Published:** 2023-01-16

**Authors:** Wenlei Zhao, Yancai Ding, Dong Chen, Yundong Xuan, Zhiqiang Chen, Xupeng Zhao, Bin Jiang, Baojun Wang, Hongzhao Li, Chengliang Yin, Xin Ma, Gang Guo, Liangyou Gu, Xu Zhang

**Affiliations:** 1Department of Urology, The Third Medical Centre, Chinese PLA General Hospital, Beijing 100853, China; 2Department of Urology, Chinese PLA NO. 942 Hospital, Yinchuan 750003, China; 3Medical School, Nankai University, Tianjin 300071, China; 4National Engineering Laboratory for Medical Big Data Application Technology, Chinese PLA General Hospital, Beijing 100853, China; 5Medical Big Data Research Center, Medical Innovation Research Division of Chinese PLA General Hospital, Beijing 100853, China

**Keywords:** kidney neoplasms, lower pole, partial nephrectomy, outcome, robotics

## Abstract

(1) Background: For completely lower pole renal tumors, we compared the perioperative outcomes of robotic partial nephrectomy via transperitoneal and retroperitoneal approaches. (2) Methods: Complete lower pole renal tumors were defined as tumors that received 1 point for the “L” element of the R.E.N.A.L. and located at the lower pole of kidney. After confirming consistency in baseline characteristics, oncological and functional benefits were compared. Pentafecta achievement was used to represent the perioperative optimal outcome, followed by multivariate analysis of factors associated with the lack of pentafecta achievement. (3) Results: Among 151 patients identified, 116 (77%) underwent robotic partial nephrectomy via a transperitoneal approach and 35 (23%) via a retroperitoneal approach. Patients undergoing transperitoneal robotic partial nephrectomy experienced more blood loss than those undergoing retroperitoneal robotic partial nephrectomy (50 mL vs. 40 mL, *p* = 0.015). No significant differences were identified for operative time (120 min vs. 120 min), ischemia time (19 min vs. 20 min), positive surgical margins (0.0% vs. 2.86%), postoperative rate of complication (12.07% vs. 5.71%). No significant differences were identified in pathologic variables, eGFR decline in postoperative 12-month (3.9% vs. 5.4%) functional follow-up. Multivariate cox analysis showed that tumor size (OR: 0.523; 95% CI: 0.371–0.736; *p* < 0.001) alone was independently correlated to the achievement of pentafecta. (4) Conclusions: For completely lower pole renal tumors, transperitoneal and retroperitoneal robotic partial nephrectomy provide similar outcomes. These two surgical approaches remain feasible options for these cases.

## 1. Introduction

Partial nephrectomy(PN) is recommended by current guidelines as the standard management for clinical T1a renal tumors [1,2], and also for patients with T1b masses when technically feasible [3]. Over the past decade, the minimally invasive approach has gained increasing acceptance and application, and robotic surgery in PN(RPN) is now a standard of treatment options [4,5]. There is still an ongoing debate about the different invasive approaches to access a renal tumor [6]. In general, transperitoneal RPN(TRPN) provides more maneuver space and avoids instrument collisions, while the retroperitoneal method(RRPN) helps expose renal hilum and avoids abdominal surgery [7,8]. This anatomically explains why the application and boundaries of PN is largely dependent on the surgeon’s experience and preferences [9]. To evaluate the two, many retrospective studies comparing the outcomes and complications of TRPN and RRPN in renal cancer cases have been reported. Zhu et al. [10], from our center, summarized the results of the research of these controls from our center. They found that cases treated by TRPN suffered significantly higher rate of minor complications, increased estimated blood loss, and longer operative time and hospital stay. The results of the included studies varied widely due to selection bias, inclusion criteria, and evaluation systems, which led to doubtful conclusions from this meta-analysis.

In order to assess treatment outcome with a common criterion, it is necessary to develop a comprehensive evaluation system. Function retention and oncologic benefit are two sides of perioperative outcomes. Hung et al. [11] defined trifecta as a ternary outcome of negative margin, no urological complications, and warm ischemia time(WIT) ≤ 25 min. With increased concern for renal function preservation, Zargar et al. [12] added “90% eGFR preservation” and “no CKD stage upgrading” to trifecta and proposed a new assessing tool of “optimal outcome”, also known as pentafecta achievement.

To obtain more statistical evidence for surgical approach choice, we examined the surgical and oncological results between TRPN and RRPN for treating completely lower pole renal masses.

## 2. Materials and Methods

### 2.1. Study Patients

With the approval of our hospital, we performed retrospective patient screening in our former established database. Only patients undergoing TRPN or RRPN between 2013 and 2016 with completely lower pole renal masses were analyzed in the present study (Appendix A). The decision of TRPN or RRPN was made by the skilled surgeons. A “skilled surgeon” was strictly defined as a surgeon who has completed more than 300 cases of laparoscopic PN and more than 100 cases of RPN. Therefore, all surgeons had been through the learning curve of TRPN and RRPN. Based on preoperative CT or MRI, two researchers separately screened kidney masses using the R.E.N.A.L. scoring system. Complete lower pole renal tumors were defined as tumors that received 1 point for the “L” element of the R.E.N.A.L. and were located at lower pole of kidney. Figure 1 presented two typical cases undergoing TRPN and RRPN, respectively. The criteria for discharge were no significant bleeding findings, removal of drains, and basic resumption of self-care.

After being screened for exclusion criteria, a total of 151 cases were enrolled, with 116 patients attributed to TRPN group and 35 patients attributed to RRPN group. This study received informed consent from each enrolled patients and obtained approval from the medical ethics committee.

### 2.2. Trocar Placement and Surgery Procedure

Trocar placement of TRPN and RRPN were performed as previously described (Appendix A) [13]. For TRPN, place the patient in a transverse cubic position of 60°–70° and support the lateral extension with a gel pad (Appendix A). After the placement of trocar and the establishment of pneumoperitoneum, the lateral peritoneum was incised along the paracolonic sulcus, and the ascending colon was pushed to the contralateral side. The surgeon cut the hepatocolonic ligament and pulled the lower edge of the liver apart, exposing the surgical area. The surgeon carefully freed the duodenum and exposed the inferior vena cava behind it, then incised the inferior vena cava sheath longitudinally. The surgeon cut the perirenal fat on the right side of the inferior vena cava, freed the renal vein and the renal artery behind it. The surgeon cut the perirenal fascia and looked for the tumor to determine the extent of resection. The surgeon used a bulldog clip to block the renal artery. The surgeon cut the renal parenchyma 0.5 cm along the outside of the tumor, and combined blunt and sharp separation until the tumor detached from the renal parenchyma, and then cleared the wound. The surgeon used a 1– Quill suture to continuously suture the wound, loosened the bulldog clamp, and closed the abdomen.

For RRPN (right side), patient was positioned in a lateral decubitus position with extended flanks (Appendix A). Insufficient flank expansion may narrow the surgical space, in which case a gel pad can be placed under the waist to raise the waist. The surgeon identified peritoneal folding, and cut the perirenal fascia and fat sac longitudinally on its medial sides. Along the surface of kidney, the surgeon combined blunt and sharp separation to expose the tumor in the gap between renal parenchyma and perirenal fat. Renal artery was found in the anterior space of the psoas muscle. The rest of the steps were similar to TRPN.

### 2.3. Study Variables and Outcomes

Patients’ baseline demographics included age, sex, body mass index (BMI), American Society of Anesthesiologists (ASA) score, Charlson comorbidity index (CCI), history of abdominal surgery, clinical presence of diabetes or hypertension, anatomical characteristics of the tumor and preoperative estimated glomerular filtration rate (eGFR). Operative variables embraced operating time, estimated blood loss, renal artery clamping time, transfusion, conversion to radical or open, surgical margin status, postoperative hospital stay, and postoperative relevant complications. The pathological slices of all subjects were re-reviewed by a skilled pathologist. We recorded complications in light of the modified Clavien–Dindo classification system [14]. eGFR before and after surgery was judged using the CKD-EPI equation [15]. We calculated the changes in eGFR of postoperative 1-day and 12-month observations. CKD upstaging was identified as a new CKD stage III–V diagnosis after surgery. After surgery, periodical follow-up was performed on each patient, and the detailed results of endpoints were noted.

### 2.4. Statistical Analyses

After the normality distribution test, for data that are skewed, the median is used to represent the characteristics of the data. Wilcoxon rank sum test, Pearson’s Chi-square and Fisher’s exact tests were applied for continuous and categorical variables, respectively. To look for potential factors for pentafecta achievement, univariate and multivariate cox regression analysis was applied. Variables involved in multivariate analysis were selected based on clinical experience and univariate analysis results. All statistical analyses were performed using R software (version 3.3.1). All tests were two sided, and *p*-values < 0.05 were considered statistically significant.

## 3. Results

A total of 116 and 35 subjects treated with TRPN and RRPN, respectively, for lower pole renal masses. Patients’ demographics and tumor features were presented in Table 1. All relevant preoperative baseline data showed similar statistical results and the differences were not statistically significant, especially for the anatomical features of tumors (*p* > 0.05 for all).

Table 2 detailed the perioperative results. Patients undergoing TRPN experienced more blood loss than those undergoing RRPN (50 mL vs. 40 mL, *p* = 0.015). Patients undergoing TRPN had similar median operating time, median ischemia time and median postoperative hospital stay, when compared with patients undergoing RRPN. The patients in the RRPN group had a lower rate of postoperative complications, either total (5.7% vs. 12.1%), minor (5.7% vs. 9.5%), or major (0.0% vs. 2.6%). All three major complications were found only in the TRPN group. One with Clavien 4 developed respiratory failure and was sent to the ICU for rescue. Two cases of Clavien 3 experienced postoperative urine leakage and had a double-J tube placement under cystoscopy. However, these differences were not statistically significant (*p* > 0.05 for all).

Pathological results and follow-up data were summarized in Table 3. Proportionally, the TRPN group had more Fuhrman grade 3–4 tumors and necrosis, but the differences were not significant between approaches in the field of other pathological features (*p* > 0.05 for all). The postoperative eGFR was similar between the two groups at day 1 (87.6 mL/min/1.73 m^2^ vs. 86.4 mL/min/1.73 m^2^) and month 12 (92.5 mL/min/1.73 m^2^ vs. 91.0 mL/min/1.73 m^2^), the proportion of eGFR decrease was also similar at day 1 (8.4% vs. 8.2%), but lower in the TRPN group at month 12 (3.9% vs. 5.4%); however, all these differences were not significant. Patients in both groups had a similar and long median follow-up time (65.2 months vs. 64.3 months, *p* = 0.341). In the follow-up period, occasional local recurrence (1/116 vs. 1/35, *p* = 0.411) and distant metastasis (2/116 vs. 1/35, *p* = 1.000) appeared in these patients of two approaches.

Pentafecta achievements between TRPN and RRPN are detailed in Table 4. Although the rate of pentafecta achievement was lower in the TRPN group than in the RRPN group, it was not significant (57.8% vs. 74.3%, *p* = 0.112). Moreover, no significant difference was found for each component (*p* > 0.05 for all). Lastly, only tumor size (OR 0.523, 95% CI 0.371–0.736, *p* < 0.001) was an independent predictor for postoperative pentafecta achievement via univariable and multivariable analyses (Table 5).

## 4. Discussion

Anatomically, PN through transperitoneal approach is easier to obtain a larger surgical field and peritoneum integrity is not required, which may significantly reduce the operation time. Just like laparoscopy, RPN was mainly performed over RRPN at first [10]. However, for lateral or posterior renal masses, TRPN can be more difficult to excise the mass and suture surgical wounds. Several centers have shared their experience with RRPN and confirmed the safety, feasibility and similar clinical outcomes compared to TRPN. As experience grows, the boundary of RRPN continues to expand, although the surgeon’s preference still plays a role. That is to say, comparative studies of different preoperative features of renal tumors were necessary to provide evidence of approach selection. In two previous studies, we have reported comparative results of RRPN and TRPN in complete endophytic and complete upper polar renal tumors, demonstrating the excellent and similar clinical outcomes and safety [13,16].

In some previous retrospective studies, the two approaches may have different baselines due to preferences [17,18], especially in the “A” domain of the RENAL score. Considering the surgeon’s general preference in the choice of surgical approach, we first validated the consistency of baseline data, including perioperative, functional, and oncological data. In terms of demography and disease characteristics, there were no significant differences between TRPN and RRPN. Notably, there was also absence of difference in the “A” part of the RENAL score between TRPN and RRPN, whether anterior (40.5% vs. 28.6%), posterior (39.7% vs. 34.3%), or not determined (19.8% vs. 37.1%, *p* = 0.106). That is, there was good comparability between TRPN and RRPN.

The surgical advantages of RRPN for cases with renal masses has been confirmed in most previous studies, such as less blood loss, shorter operating time, clamping time, and hospital stays, have been demonstrated in most previous studies, especially for lateral or posterior tumors. In the current study, we found that the differences in the incidence of postoperative complications (12.1% vs. 5.7%) and pentafecta achievement (57.8% vs. 74.3%) were noticeable but not statistically significant. Four meta-analyses have so far pooled data from RPN studies [10,19,20,21]. One of them was published by our center, in 2021, and concluded that RRPN had some advantages over TRPN in perioperative, functional, and oncological results [10]. Patients undergoing RRPN had a lower rate of Clavien–Dindo grade 1–2 complications (*p* = 0.04; OR: 1.39; 95% CI, 1.01–1.91). However, there was no significant difference in the incidence of total (*p* = 0.06; OR: 1.29; 95% CI, 0.99–1.69) and major (*p* = 0.07; OR: 0.72; 95% CI, 0.51–1.03) complications. No significant differences were found in other studies using pentafecta criteria to assess the effects of TRPN and RRPN [18,22]. It may be that the sample is not large enough.

Pathological features further confirmed the comparability between the two groups. The proportion of pathological subtypes was basically consistent with its natural incidence, and there was no significant difference. Combined with the data mentioned above, the patient’s tumor size, RENAL score, preoperative renal function and ischemia time had no significant difference, so it is logical and credible that there was an absence of difference in postoperative functional and oncological outcomes. In addition, our research reported renal function and tumor termination similarly to the previous literature [13,16].

Function retention and oncologic benefit are two sides of perioperative outcomes. Before pentafecta achievement, several studies proposed, respectively, their comprehensive evaluation concepts, or proposed improvements for existing concepts. Buffi et al. [23] proposed the Margin, Ischemia, and Complications system as their ideal outcome standard. Hung et al. [11] defined trifecta as a ternary outcome of negative margin, no urological complications, and no more than 90% loss of renal function. With increased concern for renal function preservation, Zargar et al. [12] added “90% eGFR preservation” and “no CKD stage upgrading” to trifecta and proposed a new assessing tool of “optimal outcome”, also known as pentafecta achievement. In the present study, although the percentages of pentafecta achievements varied widely (57.8% vs. 74.3%), the differences were not statistically significant (*p* = 0.112). Only decreasing tumor size was found in logistic regression analyses to be an independent risk factor associated with pentafecta achievement. The differences from surgical type were not statistically significant (*p* = 0.205), suggesting that the two approaches led to similar pentafecta achievements. Unlike the results of our previous research for completely upper pole renal tumors, RENAL score is not an independent risk factor for pentafecta achievement. Considering that the tumor size is also a component of the RENAL score, further research may be required to provide a possible explanation. In 2017, pentafecta achievement was applied in a comparison of TRPN and RRPN for large localized renal masses by Stroup et al. [18]. Multivariable analysis identified that preoperative eGFR and RENAL score were independent predictors correlated to the lack of an optimal outcome. Choi et al. [17] examined hundreds patients managed by robotic surgery and compared the value of TRPN and RRPN in treatment of localized renal mass. The results showed that baseline hemoglobin and tumor size were risk factors. SPARE, a simplified version of the PADUA scoring system, was applied by Sharma et al. [24] to predict pentafecta outcomes in patients who underwent RPN. Their study validated the SPARE scoring system in predicting pentafecta achievement in a RPN cohort and found that age, baseline eGFR, and SPARE score were risk factors for pentafecta achievement. To sum up, even with the only evaluation criteria of pentafecta outcomes, similar studies have not come to the same conclusion. This may be due to different inclusion criteria, study subjects, and surgical conditions at different centers. More studies should be conducted on the comparison between RRPN and TRPN and the factors influencing their outcomes to eliminate this discrepancy.

There were also some limitations in our research: vitally, retrospective experimental design and limited sample size. Although our comparison of preoperative data between RRPN and TRPN confirmed the comparability of the two groups, the limited sample size may still be the reason why some of the differences were not statistically significant. Moreover, given the large number of cases in our center, all the cases in the study were performed by experienced surgeons. Sufficient experience may result in some trivial differences between the two approaches being bridged. Junior surgeons should still consider the anatomic situations presented by the two approaches when interpreting these results. Despite these limitations, our comparative study remains the initial report of TRPN and RRPN in completely lower pole renal masses. The current study should be seen as a complement to our previous studies on completely endophytic and completely upper polar renal tumors [13,16]. All three studies together confirmed that RRPN and TRPN have similar and reliable benefits for renal tumors in different locations.

## 5. Conclusions

In summary, for cases with completely lower pole renal masses, TRPN and RRPN can offer similar outcomes in safety and function effectiveness. The surgical approach was not a factor in pentafecta achievement after RPN. Prospective randomized studies are needed to verify our present findings.

## Figures and Tables

**Figure 1 jcm-12-00722-f001:**
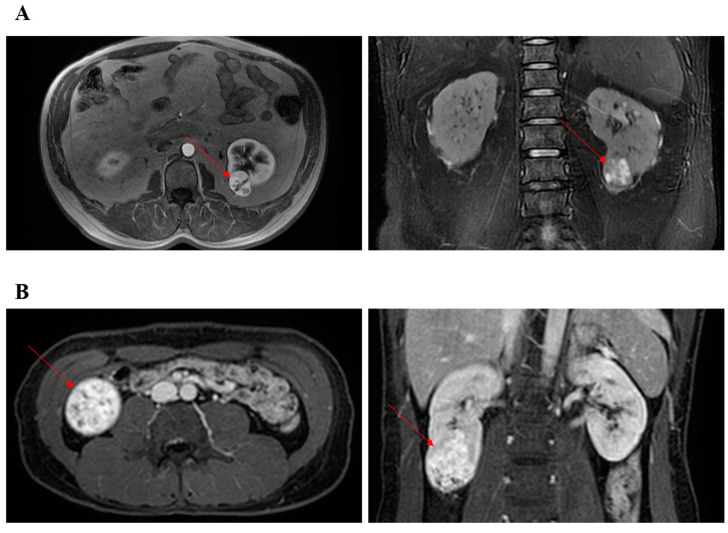
Representative images of CT or MRI. Red arrow points to renal tumor. (**A**) The patient underwent retroperitoneal robotic partial nephrectomy. (**B**) The patient underwent transperitoneal robotic partial nephrectomy.

**Table 1 jcm-12-00722-t001:** Patients’ demographics and tumor characteristics.

	Overall	TRPN	RRPN	*p* Value
No. patients	151	116	35	
Age, years, median (IQR)	52 (45–61)	53 (45–61)	51 (46–60)	0.592
Male patients, n (%)	120 (79.5)	93 (80.2)	27 (77.1)	0.811
BMI, kg/m^2^, median (IQR)	25.8 (23.7–28.1)	25.7 (23.8–28.0)	26.0 (23.6–28.4)	0.836
ASA score, n (%)				0.199
1 and 2	143 (94.7)	108 (93.1)	35 (100.0)	
3 and 4	8 (5.3)	8 (6.9)	0 (0.0)	
CCI score, n (%)				1.000
0–1	129 (85.4)	99 (85.3)	30 (85.7)	
≥2	22 (14.6)	17 (14.7)	5 (14.3)	
Clinical symptoms, n (%)	9 (6.0)	7 (6.0)	2 (5.7)	1.000
Presence of diabetes, n (%)	24 (15.9)	20 (17.2)	4 (11.4)	0.449
Presence of hypertension, n (%)	42 (27.8)	33 (28.4)	9 (25.7)	0.832
Prior abdominal surgery, n (%)	27 (17.9)	18 (15.5)	9 (25.7)	0.208
Solitary kidney, n (%)	1 (0.7)	1 (0.9)	0 (0.0)	1.000
Left tumor, n (%)	75 (49.7)	55 (47.4)	20 (57.1)	0.340
Tumor size, cm, median (IQR)	3.3 (2.5–4.3)	3.3 (2.5–4.3)	3.3 (2.5–3.8)	0.187
R.E.N.A.L. score, median (IQR)	6 (5–7)	6 (5–7)	6 (5–7)	0.310
R.E.N.A.L. complexity class				1.000
Low (4–6)	80 (53.0)	61 (52.6)	19 (54.3)	
Moderate (7–9)	71 (47.0)	55 (47.4)	16 (45.7)	
High (10–12)	0 (0.0)	0 (0.0)	0 (0.0)	
Anterior/Posterior aspect, n (%)				0.106
Anterior	57 (37.7)	47 (40.5)	10 (28.6)	
Posterior	58 (38.4)	46 (39.7)	12 (34.3)	
Not determined	36 (23.8)	23 (19.8)	13 (37.1)	
Hypothermic ischemia, n (%)	1 (0.7)	1 (0.9)	0 (0.0)	1.000
Preoperative creatinine (umol/L), median (IQR)	75.6 (63.9–85.3)	74.1 (63.7–85.2)	76.6 (67.5–85.4)	0.492
Preoperative eGFR (mL/min/1.73 m^2^), median (IQR)	96.3 (85.8–104.4)	96.2 (85.8–106.2)	96.3 (85.3–101.8)	0.785

TRPN = transperitoneal robotic partial nephrectomy; RRPN = retroperitoneal robotic partial nephrectomy; IQR = interquartile range; BMI = body mass index; ASA = American Society of Anesthesiologists; CCI = Charlson Comorbidity Index; eGFR = estimated glomerular filtration rate.

**Table 2 jcm-12-00722-t002:** Perioperative outcomes.

Variable	TRPN	RRPN	*p* Value
No. patients (%)	116 (76.8)	35 (23.2)	
Operating time, min, median (IQR)	120 (101–150)	120 (100–175)	0.710
Estimated blood loss, mL, median (IQR)	50 (50–100)	40 (20–100)	0.015
Renal artery clamping time, min, median (IQR)	19 (15–25)	20 (13–25)	0.828
Transfusion, n (%)	1 (0.9)	0 (0.0)	1.000
Conversion to radical, n (%)	0 (0.0)	0 (0.0)	1.000
Conversion to open, n (%)	0 (0.0)	0 (0.0)	1.000
Positive surgical margin, n (%)	0 (0.0)	1 (2.9)	0.232
Postoperative hospital stay, d, median (IQR)	5 (5–7)	6 (4–7)	0.735
Postoperative complications, n (%)	14 (12.1)	2 (5.7)	0.364
Minor	11 (9.5)	2 (5.7)	0.733
Clavien 1	6 (5.2)	2 (5.7)	
Clavien 2	5 (4.3)	0 (0.0)	
Major	3 (2.6)	0 (0.0)	1.000
Clavien 3	2 (1.7)	0 (0.0)	
Clavien 4	1 (0.9)	0 (0.0)	

TRPN = transperitoneal robotic partial nephrectomy; RRPN = retroperitoneal robotic partial nephrectomy; IQR = interquartile range.

**Table 3 jcm-12-00722-t003:** Pathological outcomes and follow-up data.

Variable	TRPN	RRPN	*p* Value
Tumor histology, n (%)			0.213
Clear cell RCC	103 (88.8)	29 (82.9)	
Papillary RCC	6 (5.2)	1 (2.9)	
Chromophobe RCC	5 (4.3)	2 (5.7)	
Other types	2 (1.7)	3 (8.6)	
Pathologic stage, n (%)			0.134
T1a	80 (69.0)	29 (82.9)	
T1b	36 (31.0)	6 (17.1)	
Fuhrman grade, n (%)			0.205
Low (1–2)	94 (81.0)	29 (82.9)	
High (3–4)	9 (7.8)	0 (0.0)	
Tumor necrosis, n (%)	15 (12.9)	2 (5.7)	0.362
Postoperative 1-day eGFR, mL/min/1.73 m^2^	87.6 (72.7–99.7)	86.4 (72.1–99.7)	0.930
Postoperative 1-day% eGFR decline	8.4 (2.9–16.0)	8.2 (1.8–17.7)	0.991
Postoperative 12-month eGFR, mL/min/1.73 m^2^	92.5 (79.7–101.5)	91.0 (76.9–99.7))	0.764
Postoperative 12-month% eGFR decline	3.9 (0.2–7.9)	5.4 (2.1–9.4)	0.196
Follow-up, months, median (IQR)	65.2 (54.0–76.1)	64.3 (59.0–66.6)	0.341
Oncological outcomes, n (%)			
Local recurrence	1 (0.9)	1 (2.9)	0.411
Distant metastasis	2 (1.7)	1 (2.9)	1.000

TRPN = transperitoneal robotic partial nephrectomy; RRPN = retroperitoneal robotic partial nephrectomy; RCC = renal cell carcinoma; eGFR = estimated glomerular filtration rate; IQR = interquartile range.

**Table 4 jcm-12-00722-t004:** Pentafecta analysis comparing TRPN and RRPN.

Outcome	TRPN	RRPN	*p* Value
Negative margins, n (%)	115 (99.1)	35 (100.0)	1.000
No complications, n (%)	102 (87.9)	33 (94.3)	0.733
Ischemia time ≤ 25 min, n (%)	93 (80.2)	32 (91.4)	0.136
eGFR > 90% of preop, n (%)	94 (81.0)	30 (85.7)	0.622
No CKD upstaging, n (%)	116 (100.0)	33 (94.3)	0.053
“Pentafecta”, n (%)	67 (57.8)	26 (74.3)	0.112

TRPN = transperitoneal robotic partial nephrectomy; RRPN = retroperitoneal robotic partial nephrectomy; eGFR = estimated glomerular filtration rate; CKD = chronic kidney disease.

**Table 5 jcm-12-00722-t005:** Univariable and multivariable analysis for factors associated with achieving pentafecta.

Variable	Univariate		Multivariate	
	OR (95% CI)	*p* Value	OR (95% CI)	*p* Value
Age	0.992 (0.967–1.018)	0.529		
BMI	0.919 (0.832–1.014)	0.092	0.930 (0.837–1.033)	0.176
Sex (male vs. female)	0.854 (0.375–1.943)	0.707		
Diabetes	0.568 (0.236–1.367)	0.207		
Hypertension	0.674 (0.327–1.390)	0.286		
CCI (≥2 vs. 0–1)	1.108 (0.433–2.830)	0.831		
ASA score (3 + 4 vs. 1 + 2)	0.679 (0.329–1.162)	0.109		
Prior abdominal surgery	1.074 (0.454–2.539)	0.871		
Tumor laterality (right vs. left)	1.205 (0.624–2.326)	0.579		
Tumor size	0.505 (0.359–0.710)	<0.001	0.523 (0.371–0.736)	<0.001
Preoperative eGFR	1.000 (0.978–1.023)	0.999		
RENAL score	0.888 (0.683–1.154)	0.375		
Surgical type (TRPN vs. RRPN)	2.113 (0.910–4.908)	0.082	1.779 (0.730–4.334)	0.205

OR = odd ratio; CI = confidence interval; BMI = body mass index; CCI = Charlson Comorbidity Index; ASA = American Society of Anesthesiologists; eGFR = estimated glomerular filtration rate; TRPN = transperitoneal robotic partial nephrectomy; RRPN = retroperitoneal robotic partial nephrectomy.

## Data Availability

Data are contained within the article or Appendix A.

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
