# Peer review of "Comparison of Transperitoneal and Retroperitoneal Robotic Partial Nephrectomy for Patients with Completely Lower Pole Renal Tumors"

_jcm, 2023, doi:10.3390/jcm12020722_

Round 1

Reviewer 1 Report

1.      This is an interesting study comparing two approaches for partial nephrectomy. This is a retrospective nonrandomized study and since the surgeons moved recently form transperitoneal to the less invasive retroperitoneal approach, the results may be confounded by selection bias. On the other hand, due to the learning curve, the results of the retroperitoneal approach may be affected in some degrees. These limitations should be mentioned.

2.      In the abstract and the top of the table2 as well, please provide number of the patients in each group.

3.      Please provide mean (+- Standard deviation) of the variables instead of the median.

4.      In the table 2, the complication rate is reported to be higher in all grades for TP group but there is a mismatch with the text, line 133: “The patients in the RRPN group had a lower rate of total (12.1% vs. 5.7%) and minor (9.5% vs. 5.7%) postoperative complications, a higher proportion of major postoperative complications (2.6% vs. 0.0%), however, these differences were not statistically significant (p > 0.05 for all) “

We see high grade complications in the TP group in Table 2. First, please explain in the text which complications occurred in these patents. Second, your data is in accordance with many studies that have shown that retroperitoneal approach is associated with less complications; however, you concluded that both approaches are the same. It is logical that when two approaches have similar tumor control results, the approach which has less complications is preferable. Please discuss.

5.      Table 2: Why hospital stay is long in both groups? Please explain in the method when do you decide to discharge the patients?

Reviewer 2 Report

Overall, the paper needs further refinement in terms of English use.

The introduction could potentially benefit from the following suggestions:

-        Please consider limiting the history aspect of the Introduction, as the transition from open to laparoscopic and robotic surgery is well-known.

-        I suggest that the authors focus more on the differences between trans- and retroperitoneal approaches, in terms of advantages, disadvantages and what tumoral location is favored by each access route.

-        I believe that the definition of ‘pentafecta’ is more suitable for this section.

            The Materials and methods section could be improved by:

-        Regarding the local Ethics Committee approval, it would be of great importance to mention the approval’s registration number.

-        Please consider replacing ‘skilled’ and ‘experienced’ surgeon or pathologist with more quantifiable measures, such as years of experience.

-        Although the authors have described the employed retro- and transperioneal techniques, it would be beneficial to summarize the steps for this paper as well.

-        I suggest that the authors include pictures with the trocars’ configuration, as well as relevant intraoperative steps.

-        In terms of exclusion criteria, I recommend that the authors are more precise, ideally by depicting them as a flow-chart image.

-        Finally, this section must include the technique of cold ischemia.

            The Results section could be improved by presenting the overall data in a more concise and impersonal way. Additionally, please state more clearly the median warm ischemia time, as well as the cold ischemia period.

            Lastly, the Discussions section needs precise data in order to sustain affirmations like ‘[…] so it’s logical and credible that there was absence of difference in postoperative functional and oncological outcomes.’

Round 2

Reviewer 2 Report

I would like to congratulate the authors for the current form of the manuscript. All major revision requirements have been met.